# Functional role of 18 LysR-Type transcriptional regulators of *Salmonella enterica* serovar Typhi

**Victor M. Hernández, Yitzel Gama-Martínez, Lorenzo Segovia, Ismael Hernández-Lucas** [ID] *

Departamento de Microbiología Molecular, Instituto de Biotecnología, Universidad Nacional Autonoma de México, Av. Universidad, Cuernavaca, Morelos, Mexico

* ismael.hernandez@ibt.unam.mx

## Abstract

*Salmonella enterica* serovar Typhi (*S*. Typhi) is the etiological agent of typhoid fever in humans. The genomic organization of this bacterium shows the presence of 41 LysR-Type transcriptional regulators (LTTRs). Remarkably, 23 of these LTTRs have orthologous proteins in different bacterial species where they have been characterized. However, the remaining 18 LTTRs are annotated as hypothetical proteins and their role is not described yet. In this work we report the role of these 18 proteins. We show that these transcriptional factors contain the two domains characteristic of the LTTR family. Furthermore, these LTTRs are involved in the resistance to bile salts, antimicrobial peptides, high temperature, motility, biofilm formation and porin regulation. Thus, LTTRs are essential in the *S*. Typhi response to stresses in the human hosts and in different biological process necessary for efficient infection.

## Introduction

The bacterium *S*. Typhi is a Gram-negative, rod-shaped of the *Enterobacteriaceae* family and is the etiological agent of typhoid fever. Typhoid fever can be treated with antibiotics although increasing resistance to different types of antibiotics is making treatment more complicated [1]. Therefore, understanding the different genetic strategies that *S*. Typhi uses during the infection process is of vital importance to design novel strategies to control and eradicate this disease. In this regard to establish an infection, this pathogen must survive diverse extreme conditions in the human host. To circumvent these conditions, *S*. Typhi utilizes its genetic arsenal that includes LTTRs. LTTRs are DNA-binding proteins that transcriptionally regulate genes for central metabolism, oxidative stress, porin synthesis, bile resistance, biofilm formation, pathogenesis, and other biological processes. LTTRs consist of 300–350 amino acids and contain a DNA-binding domain (DBD) at the N-terminus, a long linker helix (LH) that connects the DBD with the C-terminal periplasmic binding domain, which is also called the effector-binding domain or regulatory domain (RD). The DBD interacts

**Data availability statement:** All relevant data are within the paper and its Supporting Information files.

**Funding:** This work was supported the following grants: Dirección General de Asuntos de Personal Académico, Universidad Nacional Autónoma de México (DGAPA/UNAM, IN203621-IN202224), Ph.D Ismael Hernandez Lucas; Consejo Nacional de Humanidades Ciencia y Tecnología, (CONAHCYT, CF-2023-I-2079), Ph.D Ismael Hernandez Lucas; Programa Postdoctoral, Dirección General de Asuntos de Personal Académico, Universidad Nacional Autónoma de México (DGAPA/UNAM), Ph.D Victor M Hernández.

**Competing interests:** The authors have declared that no competing interests exist.

with the promoter and a co-inducer binds to the RD to modulate the transcriptional expression of target genes [2,3].

LTTRs are present in eukaryotic cells, archaea, and bacteria. Furthermore, these proteins are one of the most abundant transcriptional factors in bacteria [4]. *S.* Typhi contains 41 LTTRs in its genomic organization [5]. Previously, we described the function of three LTTRs in *S.* Typhi: LeuO is involved in the regulation of detoxification, porin synthesis and in the positive control of the CRISPR-Cas system [6,7]; STY0036 has a role in porin synthesis, bile resistance and in genetic transformation [8,9]; STY2660 participates in porin synthesis, bile resistance and motility [10]. These results show the relevance of LTTRs in *Salmonella* biology.

However, most of LTTRs present in *S.* Typhi are annotated as hypothetical proteins and their roles are not yet described. In this work we demonstrated that 18 previously uncharacterized *S.* Typhi LTTRs are essential in the stress responses provoked by the human host and in different biological process necessary for efficient infection.

## Materials and methods

### Bacterial strains, plasmids, and culture conditions

The bacterial strains and plasmids used in this work are listed in S1 Table. *S.* Typhi IMSS-1 (wild-type) [11] and *E. coli* strains were grown aerobically at 37°C in LB (10 g tryptone, 5 g yeast extract, and 10 g NaCl per liter), MA (7 g nutrient broth, 1 g yeast extract, 2 ml glycerol, 3.75 g $K_2HPO_4$, and 1.3 g $KH_2PO_4$ per liter [12], N-MM media [0.37 g KCl, 0.99 g $(NH_4)_2SO_4$, 0.087 g $K_2SO_4$, 0.14 g $KH_2PO_4$, 0.019 g $MgCl_2$, 1 g casamino acids, 5 ml glycerol, and 100 ml of Tris-HCl 1 M (pH 7.5) per liter] [13], or modified N-MM media [0.37 g KCl, 0.99 g $(NH_4)_2SO_4$, 0.087 g $K_2SO_4$, 0.14 g $KH_2PO_4$, 0.019 g $MgCl_2$, 1 g casamino acids, 5 ml glycerol, and 10 ml of Tris-HCl 1 M (pH 7.5) per liter]. When required, the following antibiotics were added: kanamycin (Km) 30 µg $ml^{-1}$; tetracycline (Tc) 12 µg $ml^{-1}$, and ampicillin (Ap) 200 µg $ml^{-1}$.

### DNA manipulations

Plasmid and genomic DNA isolations were carried out according to published protocols [14]. Primers for mutant construction and transcriptional fusions were provided by the Oligonucleotide Synthesis Facility at our Institute (S2 Table). Restriction enzymes, ligase, nucleotides, and polymerases were acquired from New England Biolabs, Invitrogen, or Thermo Scientific. For sequencing, double-stranded DNA was purified with the High Pure Plasmid Isolation Kit (Zymo ResearchRoche) and sequenced with an automatic Perkin Elmer/Applied Biosystems 377−18 system.

### Site-directed mutagenesis

The *Salmonella* mutants were obtained by the one-step non-polar mutagenesis procedure [15] in which the target gene was replaced with a selectable antibiotic resistance gene marker. The resistance cassette was removed using the pCP20 plasmid [16]. Each mutation was sequenced to verify the authenticity of the deletion.

## Structural analysis of LTTRs

The determined crystallographic structures of HinK, VV2_1132, CrgA, AphB, ArgP, HypT, BenM, OxyR and TsaR [17–25] were obtained from the PDB databank [26]. The structural models of the 18 LTTRs were obtained from AlphaFold Protein Structure Database (S3 Table). To analyze the structural similarity a multiple structural alignment of all these structures was built using FoldMason [27]. The Newick tree from FoldMason and the trimmed MSA were composited using iTol v7 [28].

## Construction of transcriptional reporter fusions

For transcriptional *cat* constructs, oligonucleotides (S2 Table) were designed to amplify DNA fragments of different lengths from the LTTRs regulatory regions. PCR products were double-digested with BamHI, BglII, KpnI or MluI and ligated into pKK232−8 or pKK232−9, which contain the promoterless *cat* gene. All constructs were sequenced to verify the correct DNA sequence of the PCR fragments (S1 Table).

## CAT assays

To determine the expression of the *cat* reporter gene mediated by the *S*. Typhi promoters, chloramphenicol acetyltransferase (CAT) assays were performed according to a previously published protocol [29]. Briefly, *S*. Typhi strains harboring the reporters were grown in N-MM or MA to different optical densities ($OD_{595}$). Then cells were harvested, centrifuged, washed with 0.8 ml of TDTT buffer (50 mM Tris-HCl, 30 µM DL-dithiothreitol, pH 7.8), resuspended in 0.5 ml of TDTT, and sonicated on ice for $10^{-s}$ intervals with $10^{-s}$ rest periods until the extract was clear. The homogenate was centrifuged at 12,000 g for 15 min and the supernatant used for activity measurement. For CAT assays, 5 µl of each extract were added in duplicate to a 96-well enzyme-linked immunosorbent assay (ELISA) plate, followed by the addition of 0.2 ml of a reaction mixture containing 1 mM DTNB [5,5'-dithiobis (2-nitrobenzoic acid)], 0.1 mM acetyl-coenzyme A (acetyl-CoA), and 0.1 mM chloramphenicol in 0.1 M Tris-HCl, pH 7.8. The absorbance at 412 nm was measured every 5 s for 5 min using a Ceres 900 scanning auto reader and microplate workstation. The protein concentration of the cell extracts was obtained using the bicinchoninic acid (BCA) protein assay reagent (Pierce). Protein values and the mean rate of product formation by CAT were used to determine CAT-specific activity as micromoles per minute per milligram of protein.

## Growth curves in Ox-bile and sodium deoxycholate

The *S*. Typhi wild-type and mutant strains were grown for 24 h on LB plates at 37°C. A bacterial colony was inoculated in liquid LB broth (5 ml) and grown for 16 h at 37°C, 200 rpm. Then, 50 ml of LB broth supplemented with 7.5% bile salts (Sigma Chemical, St. Louis, MO) or 5% sodium deoxycholate were inoculated independently to an $OD_{595}$ of 0.01. The cultures were incubated at 37°C, 200 rpm during 12 h with measurements being done every 2 h.

## Microtiter plate biofilm formation assay

The quantification of biofilm formation was performed according to a previously established protocol [30]. Briefly, bacterial cells were grown overnight in LB broth (5 ml) at 37°C, 200 rpm. Cells were diluted 1:100 in fresh LB without NaCl to stimulate biofilm production. Two hundred microliters of this dilution were added per well in a 96-well polystyrene microtiter plate (Costar Cat. No. 3599, flat bottom with lid). Four replicate wells were prepared for each strain. Microtiter plates were incubated at 30°C for 24 h. Total bacterial growth was measured at $OD_{600}$, using a GloMax®-Multi Detection System (Promega). The planktonic cells were then discarded and the plates washed three times with water. The remaining biofilm was fixed with 200 µl per well of methanol (100%) and stained with a 0.2% solution of crystal violet diluted in water. After incubation at room temperature for 10 min, the plates were rinsed three times with water. The dye was solubilized by adding 200 µl of 33% acetic acid to each well and incubating the plate at room temperature for 15 min. The $OD_{560}$ was determined with the microplate reader. The amount of biofilm formed is reported as the ratio of the $OD_{560}/OD_{600}$ values [31].

## Preparation of outer membrane proteins

Outer Membrane Proteins (OMPs) were isolated from *S.* Typhi IMSS-1 and its derivative mutant strains grown in N-MM to an $OD_{595}$ of 1, according to previous protocols [32]. Fifteen milliliters of each culture were harvested and centrifuged at 6,000 g for 10 min at 4°C. Cells were resuspended in 500 µl of 10 mM $Na_2HPO_4$ buffer (pH 7.2) and sonicated on ice until the suspensions were clear. Intact cells and debris were eliminated by centrifugation (3,500 g) for 2 min, and the supernatants were transferred to clean microcentrifuge tubes and membrane fractions were pelleted by centrifugation at 13,500 g for 1 h at 4°C. Inner membrane proteins were solubilized by resuspension in 500 µl of 10 mM $Na_2HPO_4$ buffer, pH 7.2, containing 2% Triton X-100 for 30 min at 37°C. After incubation, the samples were centrifuged at 13,500 g for 1 h at 4°C. The remaining outer membrane insoluble fraction was washed with 500 µl of 10 mM $Na_2HPO_4$, pH 7.2, centrifuged at 13,500 g for 1 h at 4°C, and finally resuspended in 50 µl 1X PBS, pH 7.4. OMP concentrations were determined by BCA assay, and 15 µg of each sample was analyzed by SDS-12% polyacrylamide gel electrophoresis. One-dimensional OMPs gels were visualized by staining with Coomassie brilliant blue.

## Swimming assays

To evaluate the motility of *S.* Typhi wild-type and its derivative mutant strains, the cells were grown aerobically at 37°C to the mid-logarithmic phase (1.0 $OD_{595}$) in 50 mL of N-MM. Five microliters of culture were spotted gently in the middle of a swim plate (N-MM, 0.3% bacteriological agar). The plates were incubated at 37°C for 12 h. The rates of migration from the point of inoculation, visible as a turbid zone, were measured at 12 h. The results are the means of at least 3 independent experiments.

## Growth curves at 41.5°C

*S.* Typhi wild-type and the mutant strains were grown 24 h in LB plates at 37°C. A bacterial colony was inoculated in liquid LB broth (5 ml) and grown for 16 h at 37°C, 200 rpm. Then, 50 ml of modified N-MM were inoculated independently with the pre-inoculum to an initial $OD_{595}$ of 0.1. The cultures were incubated at 41.5°C, 200 rpm during 12 h with $OD_{595}$ measurements being done every 2 h.

## Growth curves with protamine

*S.* Typhi wild-type and the mutant strains were grown 24 h in LB plates at 37°C. A bacterial colony was inoculated in liquid LB broth (5 ml) and grown for 16 h at 37°C, 200 rpm. Then, 50 ml of LB broth supplemented with 0.4 mg/mL of protamine sulfate (Sigma Chemical, St. Louis, MO) were inoculated independently with the pre-inoculum to an initial $OD_{595}$ of 0.02. The cultures were incubated at 37°C, 200 rpm for 12 h, recovering 1 ml of culture every 2 h, which was washed and resuspended in 1 ml of 1X PBS to determine its $OD_{595}$.

# Results

## LTTR identification, domain organization, structural determination and functionality in *S.* Typhi

The BioCyc database [33] shows the presence of 41 LTTRs in the *S.* Typhi CT18 genome [5]. To corroborate that these proteins had the characteristic LysR-type winged helix-turn-helix DNA-binding domain or DBD, as well as the regulatory domain or RD, the Supefamily 2.0 database [34,35] was used. The data obtained confirmed the presence of 41 LTTRs in the *S.* Typhi genome [5]. This LTTR collection shows the presence of 16 hypothetical proteins without assignment function as well as of STY0036 [8] and STY2660 [10]. The structural model of these proteins was compared with the functionally characterized structures of HinK, VV2_1132, CrgA, AphB, ArgP, HypT, BenM, OxyR and TsaR [17–25]. This analysis was performed using the FoldMason program [27] to identify the structural similarity and infer the possible function of the 18 LTTRs. The results of structural alignment showed an identity of 13–66% for the complete polypeptides. The identity for

the N-terminal domains ranged from 17 to 83%. The linker-helix domains showed an identity of 5–82%, and the C-terminal domains shared an identity of 9–61%.

On the basis of the structural alignment, four groups were generated. Group A consists of HinK, TsaR, STY0651, STY0036, STY3415, STY1693, VV2_1132, STY3165, and STY1537. The HinK regulator is involved in regulation of the type III secretion system, chemotaxis, swarming motility, and virulence [36,25]. TsaR participates in the degradation of *p*-toluenesulfonate [37] and STY0036 is related to bile tolerance [8], while the remaining members have not been characterized. Group B consists of STY0341, STY0159, STY4196, CrgA, STY0277, AphB and STY3547. CrgA regulates swimming motility and virulence [38]. AphB is involved in acid and oxidative stress, swimming motility, adhesion, porin and toxin regulation, pili formation, biofilm, and virulence [39–44], while the remaining members have not been characterized. Group C consist of ArgP, HypT, STY2821, and STY2510. ArgP is involved in the biosynthesis and transport of lysine, as well as in arginine transport [45–47]. HypT is involved in the response to oxidative stress [48]. STY2821, and STY2510 have not been characterized until now. Group D is formed by OxyR, BenM, STY1578, STY2660, STY0730, STY3158, and STY4468. OxyR is involved in the response to oxidative stress, biofilm formation, swarming and swimming motility, iron metabolism, quorum-sensing, protein synthesis, and oxidative phosphorylation [49–51]. BenM regulates the expression of genes involved in the degradation of aromatic compounds [52] and STY2660 is involved in bile tolerance and swimming motility [10], while the remaining members have not been characterized (Fig 1).

The structural data generated in this work shows the presence of four structural LTTR groups, and the literature of some of them such as STY0036, STY2660, CrgA, AphB and OxyR demonstrated that these LTTRs are involved in motility, biofilm formation, porin regulation and bile salts resistance [8,10,38,44,50].

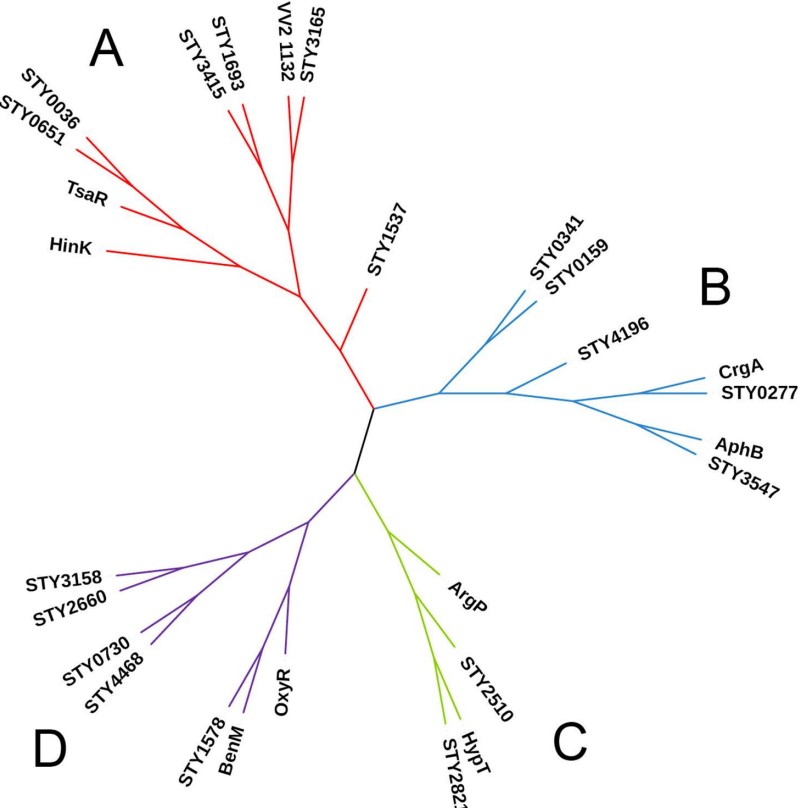

**Fig. 1. Structural clustering of 18 LTTRs of *S.* Typhi IMSS-1 using the FoldMason program.**

Based on these data we decided evaluated the mentioned phenotypes, and additional human stress conditions faced by *S.* Typhi in the infection process, as well as inducing conditions of *Salmonella* pathogenic islands I and II of the 18 LTTRs described here.

Thus, initially we evaluate at the transcriptional level the genetic expression of the LTTRs described in this work. Therefore, the regulatory region of each of the 18 LTTRs was fused to the *cat* reporter gene to obtain 18 individual LTTR transcriptional fusions, and transcriptional experiments in a rich medium A (MA) [12] and in N-minimal medium (N-MM) were performed. In the rich medium several genes of the *Salmonella* Pathogenicity Island-I were expressed. In N-MM, which is deficient in magnesium and phosphate and contains glycerol as carbon source, and several *Salmonella* Pathogenicity Island-II genes were induced in this condition [13]. The evaluation in these conditions is an indirect way of knowing whether these LTTRs have any role in similar conditions found within the human host. The expression results shows that the 18 LTTRs display different transcriptional activity values in N-MM, while in rich medium low or null expression of these LTTRs was found (Fig 2).

These results support that these LTTRs are not induced as *Salmonella* Pathogenicity Island-I genes and most of them are expressed differentially in N-MM. This suggests their role in growth under this condition. To evaluate whether these LTTRs are involved in the regulating growth in N-MM, a collection of 18 individual LTTR mutants was made by the Datsenko and Wanner method. The growth kinetics of the mutants show similar optical density ($OD_{595}$) over 12 h in the wild-type strain and in the 18 LTTR mutants (S1 Fig). Therefore, these LTTRs are not involved in *S.* Typhi growth in N-MM.

### *S.* Typhi LTTRs role under stress conditions

**LTTRs are involved in bile resistance.** *S.* Typhi is a pathogen that causes typhoid fever in humans. To stablish an infection in the host, this bacterium must survive the presence of bile salts in the gut and gallbladder. To evaluate the role of LTTRs in bile resistance, growth rate experiments were performed in LB in the presence of Ox-bile. Previous results from our laboratory showed that the wild-type *S.* Typhi strain is able to survive in the presence of 7.5% of Ox-bile [6], so we evaluated the 18 LTTR mutants in the presence of this concentration of Ox-bile. The results show three different

**Expression profile of LTTRs in *S.* Typhi**

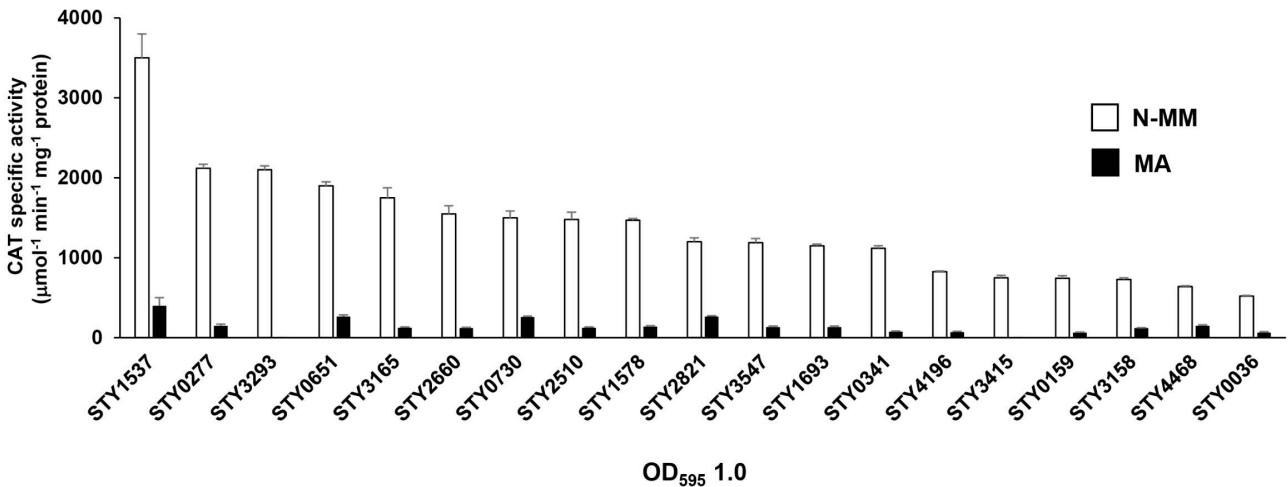

**Fig 2. Transcriptional profile of 18 LTTRs of *S.* Typhi IMSS-1 in N-MN and MA.**

growth kinetics in the presence of 7.5% Ox-bile. The first phenotype corresponds to the LTTRs mutants STY0277 and STY0159, which have a delay in the exponential growth phase, with at 8 h $OD_{595}$ values of 0.295 and 0.214, respectively, versus 0.828 for the wild-type IMSS-1. These mutants after 12 h show similar $OD_{595}$ as the wild-type strain (Fig 3A). The second phenotype was shown by LTTR mutants STY4196, STY0341, STY1578, STY1693, STY2821 and STY2510, with an average $OD_{595}$ of only 0.080–0.196 at 8 h. These mutant strains did not attain the $OD_{595}$ of the wild-type strain after 12 h (0.878) (Fig 3B). The third phenotype corresponded to the LTTR mutants STY0651, STY3547, STY0036, STY1537, STY0730, STY2660, STY3158, STY3165, STY3293, STY3415 and STY4468. These strains were unable to growth in the presence of Ox-bile 7.5% (Fig 3C).

To further confirm these results the same bacterial strains were grown in the bile salt sodium deoxycholate (DOC), a highly abundant bile salt present in the human gallbladder [53]. Previously we showed that the wild-type *S.* Typhi grows in the presence of 5% DOC [9]. Therefore, the mutant and wild-type strains were evaluated in 5% DOC.Two phenotypes were observed in this condition. In the first, LTTR mutants STY1693, STY0651, STY0730, STY2821, STY4196, STY2660, STY0277, STY2510 and STY3547 had an average $OD_{595}$ of 0.214–0.486 after twelve hours, while wild-type strain IMSS-1 had an $OD_{595}$ of 0.814 (Fig 4A). The second phenotype corresponded to the LTTR mutants STY4468, STY1578,

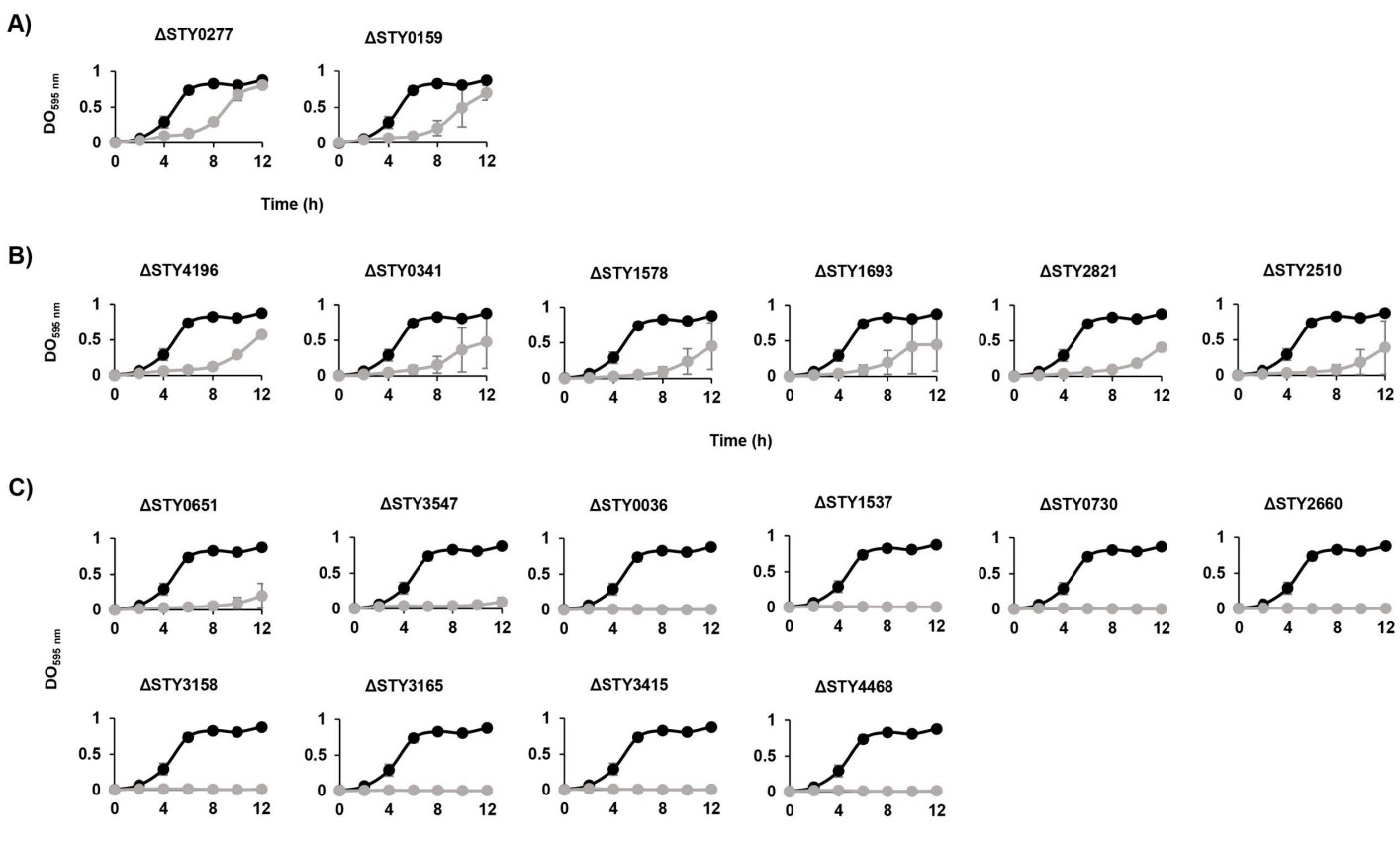

**Fig 3. Growth curves of *S.* Typhi IMSS-1 wild-type (•) and LTTR mutants (◦) in LB supplemented with 7.5% Ox-bile.** (A) LTTRs mutants that showed a delay in the exponential growth phase but reached an OD comparable to the wild-type strain. (B) LTTRs mutants that exhibited reduced growth in comparison to the wild-type strain. (C) LTTR mutants unable to growth in the presence of Ox-bile.

**Growth curves in LB supplement with sodium deoxycholate 5 %**

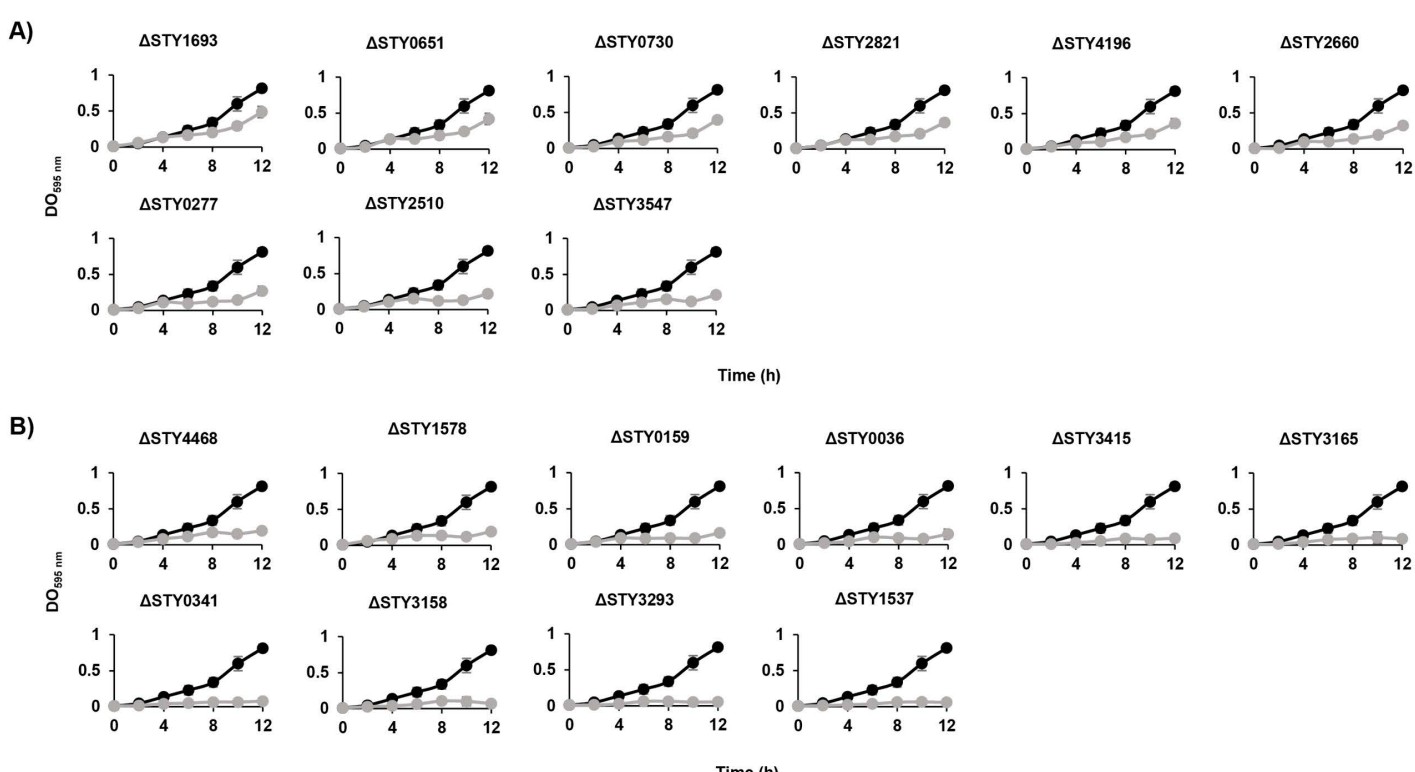

**Fig 4. Growth curves of *S*. Typhi IMSS-1 wild-type (•) and LTTR mutant (∘) in LB supplemented with 5% sodium deoxycholate. (A)** LTTRs mutant strains that exhibited reduced growth in comparison to the wild-type strain. **(B)** LTTR mutant strains unable to grow in the present of DOC.

STY0159, STY0036, STY3415, STY3165, STY0341, STY3158, STY3158 and STY1537. These strains had an $OD_{595}$ of <0.200 after 12 h in LB supplemented with 5% DOC (Fig 4B). Thus, LTTRs are involved in bile and DOC resistance to distinct extents.

**LTTR are involved in biofilm formation.** Bile salts are stored in the human gallbladder, where gallstones can develop. *S*. Typhi is able to form biofilms on these hardened structures [54]. Therefore, we evaluated biofilm formation in the wild-type *S*. Typhi and mutant strains. The results show that the wild-type *S*. Typhi induces low levels of biofilm, as measured using crystal violet detection. Two phenotypes were observed for the mutant strains. The first phenotype corresponded to LTTR mutants STY0159, STY4196, STY4468 and STY3415, which show biofilm formation similar as to the wild-type. The second phenotype was shown by the LTTR mutants STY3547, STY0277, STY2821, STY0651, STY0036, STY0341, STY2660, STY2510, STY3158, STY0730, STY1578, STY3165, STY1537 and STY1693 that present biofilm values of 0.565–0.760, and the wild-type present values of 0.165 (Fig 5). Thus, these mutant strains display of 3.42 to 4.60-fold higher than the wild-type strain.

**LTTR are involved in swimming motility.** In the biofilm formed in gallstones, *S*. Typhi remains static. However, this bacterium is able to move from this structure to the small and large intestine, and is finally released to the environment to initiate a new infection process. Motility is an essential feature of *S*. Typhi for an efficient dissemination to the environment. Thus, we evaluate the swimming motility of the wild-type strain and the mutant strains in semi-solid N-MM (low agar concentration). The results showed that the wild-type strain display a migration rate of of 2.87 cm away from the

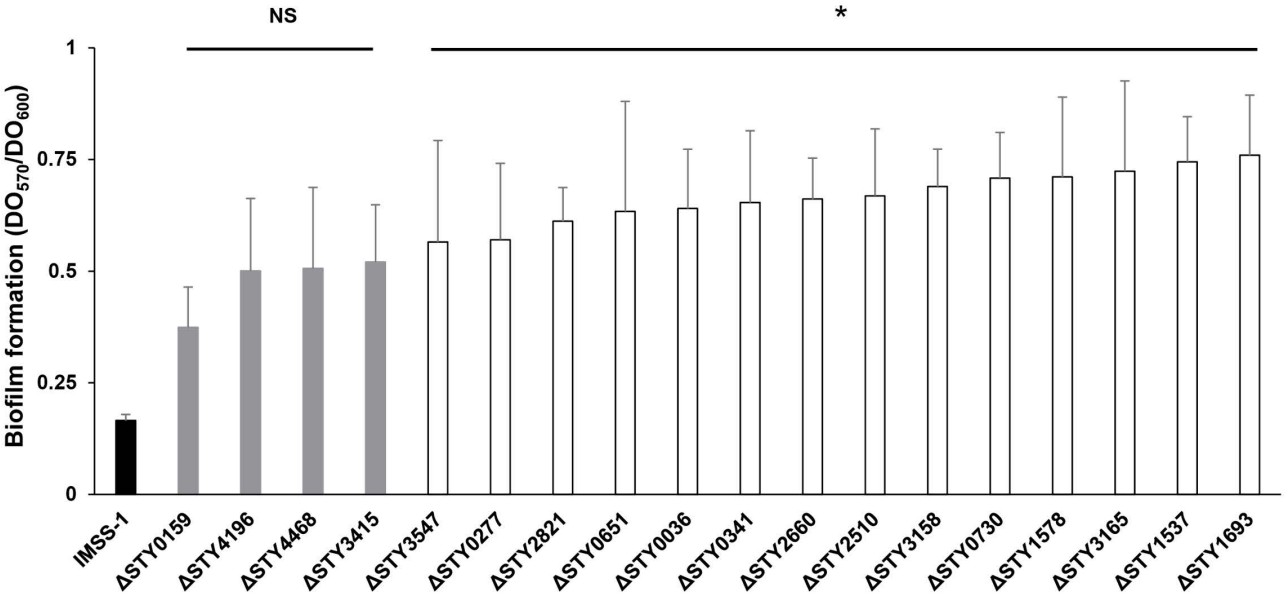

**Fig 5. Biofilm formation of *S.* Typhi IMSS-1 wild-type and LTTR mutants in LB without NaCl.** The black bar represents the IMSS-1 wild-type strain, the gray bars represent LTTR mutants with no statistically significant differences (NS) compared to the wild-type. The white bars represent LTTR mutants with statistically significant difference (* = $p < 0.05$), the differences ranging between 3.42 to 4.60-fold more. Values represent the mean ± standard deviation of three independent experiments.

semi-solid stab, while the LTTR mutant strains STY0159 and STY2510 had an average diameter of 4.0 cm, each mutant, with no significative differences in motility with wild-type (Fig 6).

The mutant strains STY2660, STY1537, STY3158, STY0730, STY2821 and STY4196 had an average mobility diameter of 4.40–4.60 cm, mutants STY3165, STY1693 and STY4468 they display an average in motility diameter of 4.63–4.83 cm, and STY0341, STY0651, STY1578, STY3415, STY0036, STY3547 and STY0277 had diameters of 4.9 to 5.1 cm. Thus, the first group of mutants did not show significant motility differences from the wild-type (Fig 6). However, the second, third and the fourth group of mutants had 1.54–1.60, 1.63–1.68 and 1.71–1.78 higher motility diameters than the wild-type strains, demonstrating the LTTRs are involved in *S.* Typhi motility to different extents.

**LTTR are involved in temperature resistance.** A distinctive feature of *S.* Tyhi in the infection process is the induction of fever in humans, thus these bacteria need to survive high temperatures in the host. *S.* Typhi wild-type strain IMSS-1 is able to growth in temperatures from 4°C to 41.5°C. Thus, the LTTR mutant strains were evaluated in modified N-MM at 41.5°C. Growth rate experiments demonstrated that the mutant strains STY0277, STY0341, STY0651, STY0159, STY4468, STY4196, STY3547, STY2821, STY1578, STY2510 and STY3415 showed an $OD_{595}$ similar to the wild-type strain ($DO_{595}$ 0.645) (Fig 7A). The mutant strains STY0036, STY0730, STY1693, STY1537, STY3158 and STY3165 had an $OD_{595}$ of 0.429–0.482 at 12 h, however the wild-type display an $OD_{595}$ of 0.645. Finally, LTTR mutant STY2660 showed an $OD_{595}$ of 0.334 in comparison with wild-type that present an $DO_{595}$ 0.645 (Fig 7C). Thus, three phenotypes were observed, one set of bacterial strains that growth as the wild type strain, a second group of mutants that display a slightly decrease in the growth rate and the STY2660 that growth 50% less compared with the wild type strain. These results show that more genetics elements than LTTRs are involved in temperature resistance, or maybe obtain double or triple LTTR mutants including the STY2660 deficient strain would be a nice strategy to determine whether LTTRs are essential in temperature resistance.

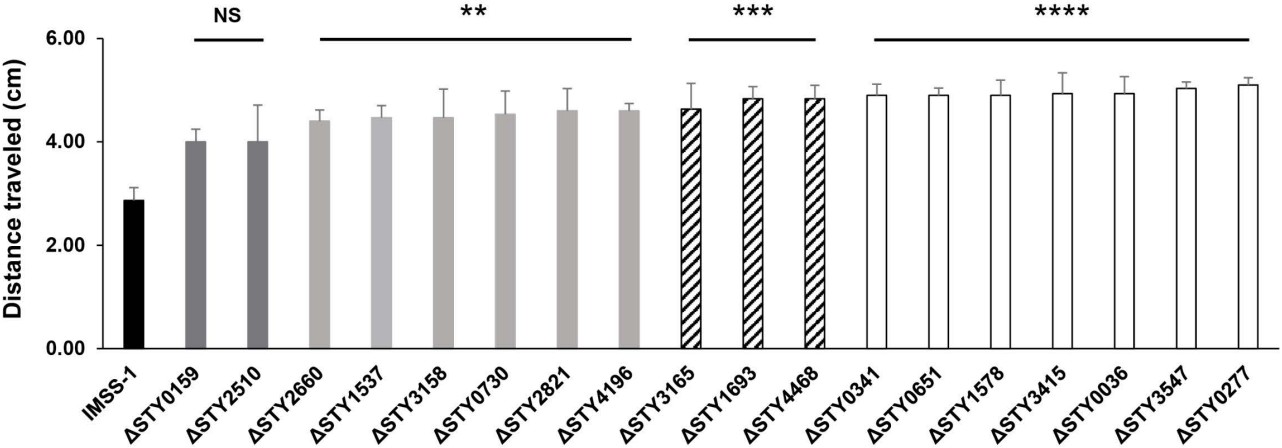

**Fig 6. Swimming motility behavior of *S.* Typhi IMSS-1 wild-type (black bar) and their derivative LTTR mutant strains (dark gray, light gray, striped and white bars) in N-MM (0.3% agar) incubated at 37°C for 12 h.** NS = statistically not significant. Statistically significant difference ** = $p < 0.01$; *** = $p < 0.001$; **** = $p < 0.0001$.

## LTTRs are involved in antimicrobial cationic peptide resistance and cell envelope structuration

**LTTRs are involved in antimicrobial resistance.** The increased incidence of multi-drug-resistant *Salmonella* has become a major global health concern. The emergence of antibiotic-resistant bacterial strains has led to an increased attempt to develop alternative antimicrobial agents. Among these, protamine, a naturally occurring polypeptide, has been investigated for its ability to inhibit the growth of various pathogenic bacteria. Furthermore, this compound has been studied as a natural preservative with the potential to prevent *Salmonella* growth in food products [55]. In this work the role of LTTRs in protamine resistance was evaluated and the results shows that the wild-type *S.* Typhi is able to growth in LB medium in the presence of 0.1 to 0.4 mg/mL of protamine sulfate. Growth curve experiments in 0.4 mg/ml protamine demonstrated that mutant strains STY0036, STY0277, STY0341, STY0651, STY0159, STY4468, STY4196, STY3547, STY2821, STY1578, STY2510 and STY3415 grew similarly to the wild-type strain ($OD_{595}$ 1.157). In contrast, LTTR mutants STY3547, STY4468, STY4196 and STY3415 had a reduced growth rate $OD_{595}$ of 0.437–0.519, compared with the wild-type *S.* Typhi strain $OD_{595}$ of 1.157 in LB supplement 0.4 mg/ml of protamine (Fig 8).

**LTTR are involved in the transcriptional regulation of outer membrane proteins.** Outer membrane proteins are fundamental for antimicrobial resistance in gram-negative bacteria [56]. Thus, we evaluated the porin profile of the wild-type strain and LTTR mutant strains grown in N-MM. The results showed that the LTTR mutants STY0159, STY0341, STY0651, STY1578, STY1693, STY2510 and STY2821 are devoid of OmpF. In the mutants STY0277, STY3415, STY3547, STY4196 and STY4468 OmpC was absent. Both OmpC and OmpF were absent in STY0036, STY730, STY1537, STY2660, STY3158 and STY3165 (Fig 9). These results illustrated that all the LTTRs evaluated in this work have a role in porin synthesis. Thus, is fundamental to evaluate whether LTTRs regulate directly or indirectly porin synthesis. In this respect previous work by our group has focused on the *ompC* and *ompF* regulation mediated by LTTRs, and the results demonstrate that the LTTRs STY2660 and STY0036 described in this work, directly control the expression of the master porin regulator *ompR* and this protein directly induces the expression of both *ompC* and *ompF*. Thus, this LTTRs are pivotal indirectly in the *ompC* and *ompF* genetic expression in *S.* Typhi [8,10].

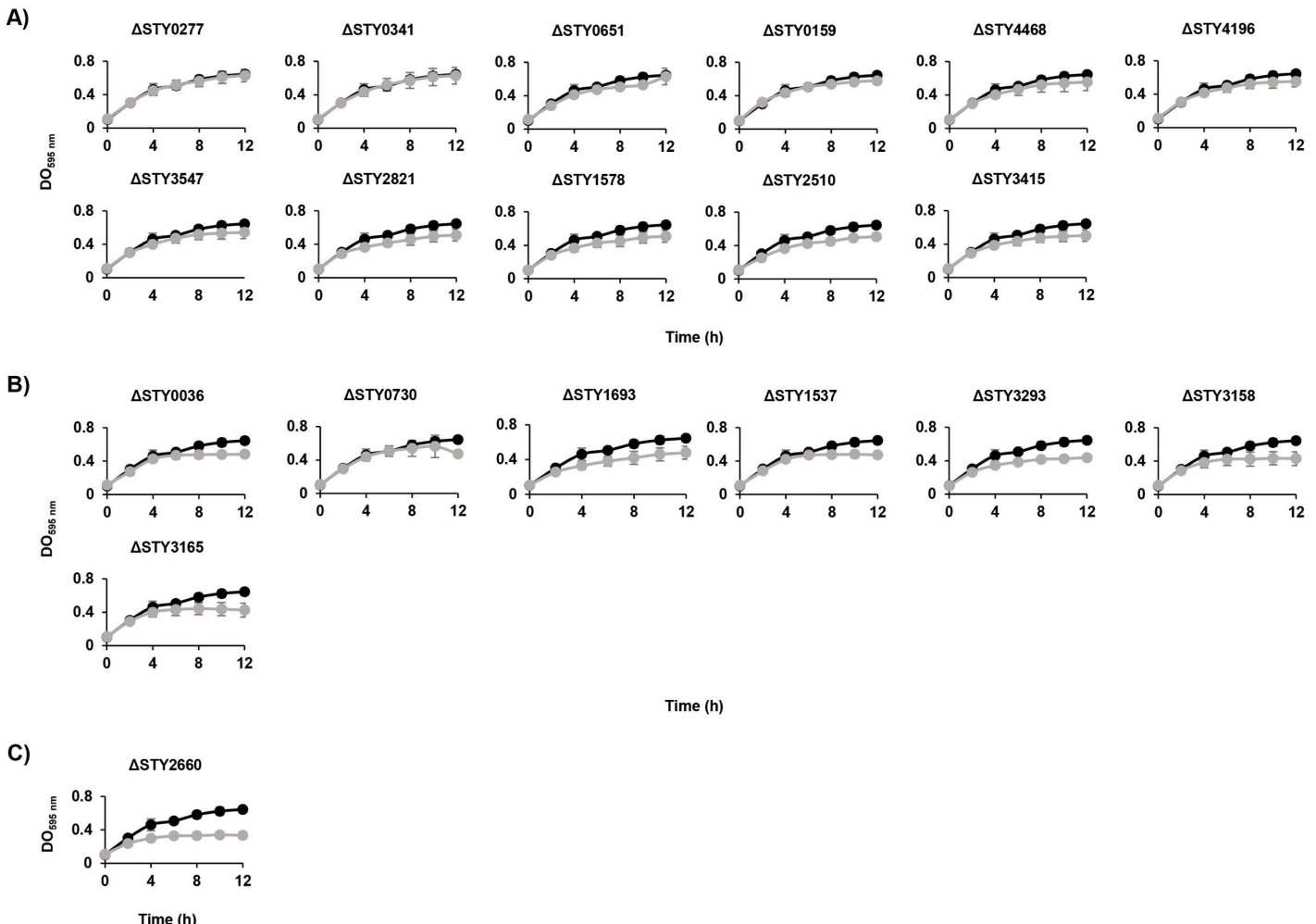

**Fig 7. Growth curves of *S*. Typhi IMSS-1(•) and LTTR mutants (•) strains in modified N-MM at 41.5°C. (A)** LTTR mutant strains showing similar growth to the wild-type strain. **(B)** LTTR mutant strains that exhibited reduced growth ($DO_{595}$ 0.429-0.482) compared to the wild-type strain ($DO_{595}$ 0.645). **(C)** LTTR mutant strain that present a decrease in growth rate ($DO_{595}$ 0.334) compared to the wild-type strain ($DO_{595}$ 0.645).

## Discussion

In this work an *in silico* structural determination and the role of 18 LTTRs of *S*. Typhi in stress conditions similar to those in human host. The data obtained shows that all the LTTRs evaluated contain the 3D structural characteristic DNA-binding domain (DBD) and an induced binding domain (IBD) and the identity between the complete 3D structures of the 18 LTTRs ranged from 13 to 66%. This data support that these 18 hypothetical proteins are LTTRs. The experimental results shows that the 18 LTTRs are divided in four functional groups according to their role in different stress conditions (Fig 10).

The functional analysis shows that in group I consists of the wild-type strain, which is able to grow in Ox bile, DOC, protamine, and at a temperature of 41.5°C. Its swimming motility and biofilm formation are limited and it produces the OmpC and OmpF porins (Fig 10).

**Growth curves in LB supplement with protamine 0.4 mg/mL**

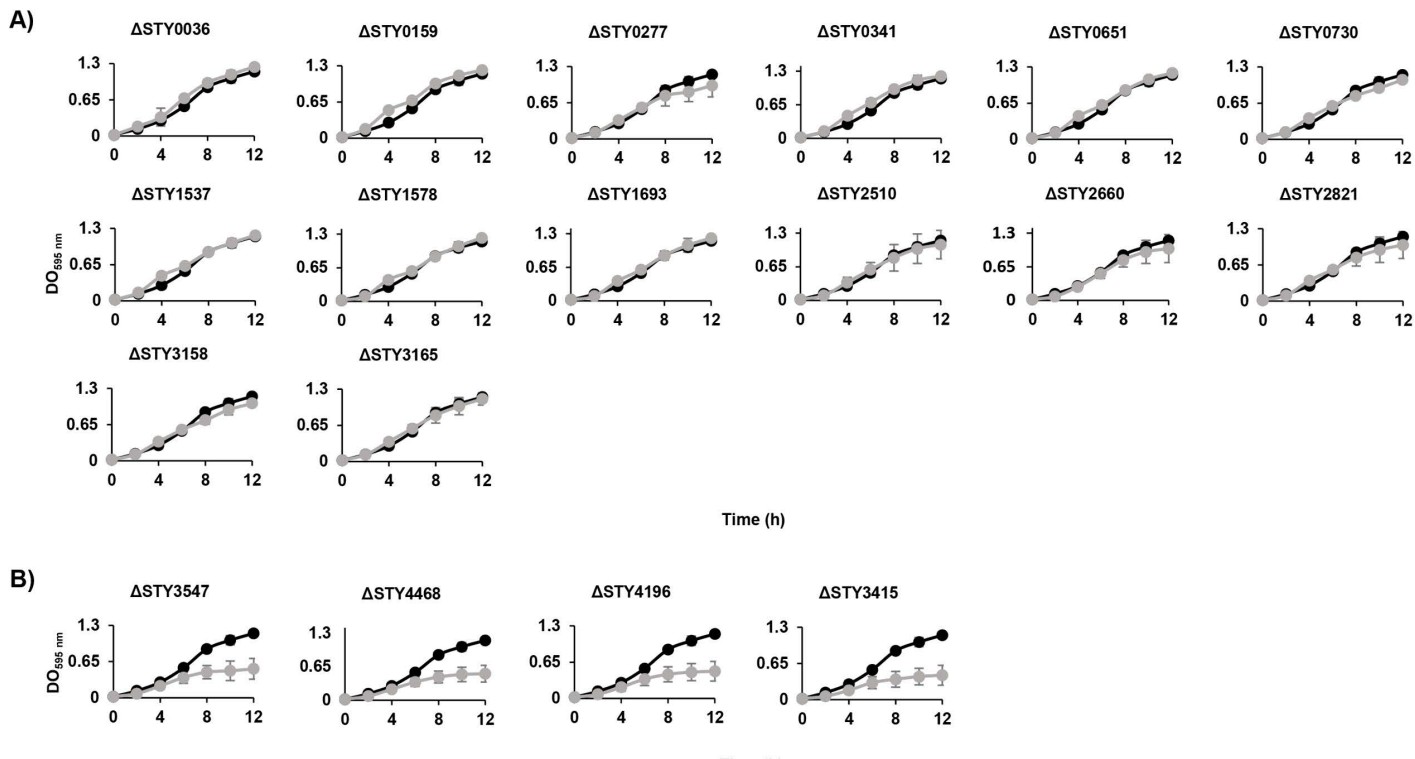

**Fig 8. Growth curves of *S*. Typhi IMSS-1 wild-type (•) and LTTR mutants (◦) in LB supplemented with 0.4 mg/mL protamine sulfate at 37°C. (A)** LTTRs mutants that showed growth comparable to the wild-type strain. **(B)** LTTRs mutants that exhibited reduced growth in comparison to the wild-type strain.

**Porin profile in N-MM at 37°C.**

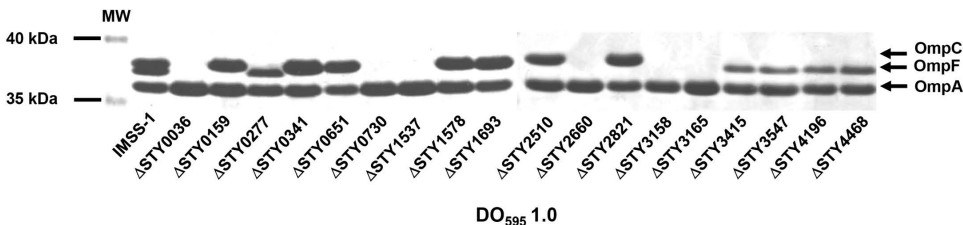

**Fig 9. Electrophoretic pattern of Coomassie brilliant blue-stained outer membrane proteins, separated by 0.1% SDS-15% PAGE of *S*.** Typhi IMSS-1 wild-type and their derivative LTTR mutants grown in N-MM to an $OD_{595}$ of 1.0 at 37°C. The major OMPs: OmpC, OmpF, and OmpA are indicated with a black arrow.

In Group II, there are LTTR mutants that grow like the wild-type in the presence of Ox-bile and mutants unable to grow in this condition. In the presence of DOC and at 41.5°C the growth of most of the LTTR mutants was reduced. In protamine, group II mutant growth decreased by 50%. The motility and biofilm formation by this group were higher than those of the wild-type strain. This group produced OmpF but not OmpC (Fig 10).

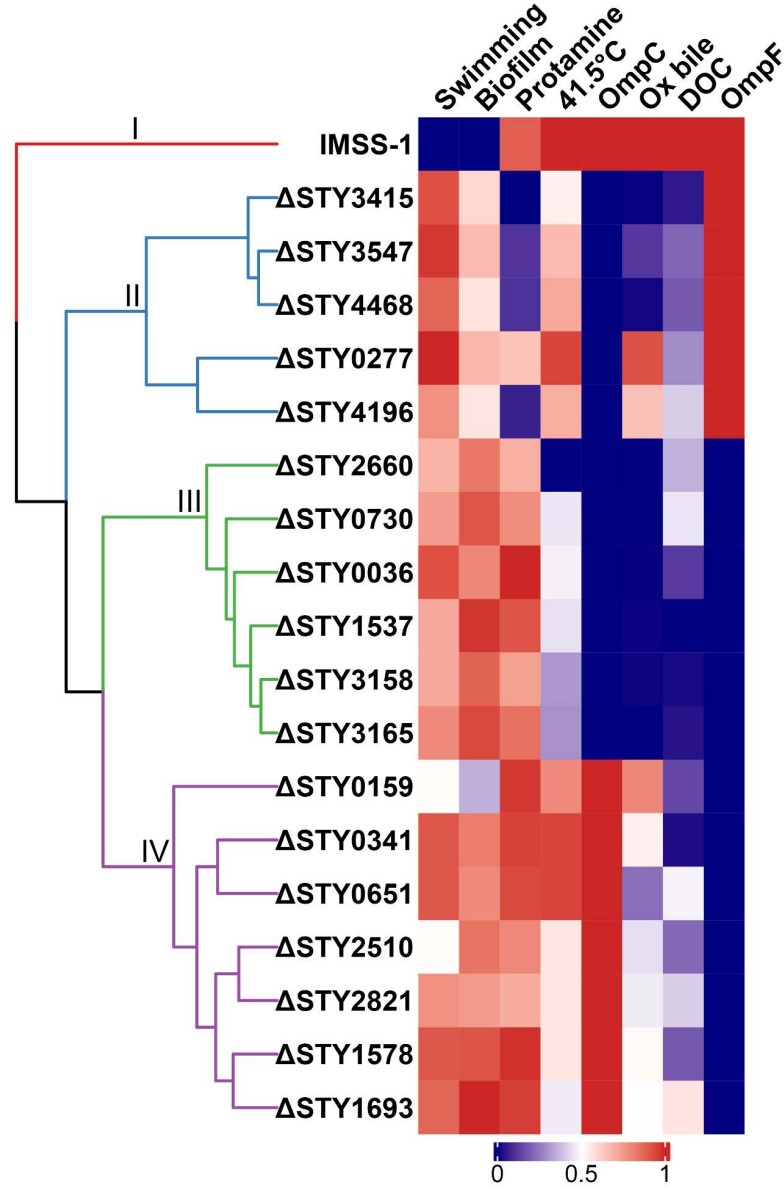

**Fig. 10. Functional groups of *S*. Typhi IMSS1 wild-type strain and LTTR mutants.** The dendrogram allowed us to identify four functional groups (I–IV) and the color in the heatmap represents the determined phenotype.

Ox-bile, DOC and growth at 41.5°C inhibited the growth of mutants in group III, while in protamine they grew at a rate similar to the wild-type strain. The motility and biofilm formation values in this group were higher than that of the wild-type. All of the LTTR mutants in this group lack OmpC and OmpF (Fig 10).

In group IV, the presence the Ox-bile, DOC, 41.5°C and protamine slightly slowed their growth. The motility and biofilm formation by this group were higher than in the wild-type strain. These mutants produced the OmpC but not the OmpF porin (Fig 10).

Overall, these results highlight the differential roles of LTTRs under the tested stress conditions. To further explore their potential involvement in virulence regulation, we examined published data on one of these regulators, STY0341.

The LTTR STY0341 characterized in this study is homologous to SinR from *Salmonella* Typhimurium. Previous studies have shown that *sinR* mutants are attenuated in their ability to replicate within macrophages, and that *sinR* expression is regulated by HilD, a major transcriptional activator involved in *Salmonella* invasion of intestinal epithelial cells [57,58]. These findings suggest that STY0341 could also play a role in *Salmonella* Typhi virulence. Furthermore, *in silico* analysis using STRING [59] revealed that STY0341 is co-expressed with HypT, a protein involved in oxidative stress resistance in *S.* Typhimurium. In contrast, the STRING analysis of the remaining 17 LTTRs showed no co-expression with known virulence genes or with transcriptional regulators associated with *Salmonella*'s free-living state.

It is relevant to mention that *S.* Typhi presents one of the highest numbers of LTTRs in the *Salmonella* genus [5]. Like many bacterial transcriptional regulators, LTTRs originated by gene duplication, and the resulting multiple LTTRs are partially redundant. This is of great relevance to organismal evolution, since it protects the regulators from potentially harmful mutations, and maintain a pool of functionally similar, yet diverse gene products. This produces a source of evolutionary novelty at the transcriptional level.

The *S.* Typhi LTTR duplications are specialized in regulating specific process such as growth in ox-bile, biofilm formation, porin synthesis and motility, suggesting that the bacteria use these multiple LTTRs to ensure the required performance of these biological activities. In this regard there are many biological processes that use multiple genes originating from gene duplications to perform adequately. For instance, 16s rRNA genes or ribosomal proteins are present in multiple copies in many organisms to ensure the correct performance of protein synthesis, a fundamental phenomenon in all living organisms. Thus, LTTR gene duplications and their role in multiple biological process is a digital printing of their relevance in *Salmonella* biology.

There are other examples of the role of multiple LTTRs in specific process. In *Aeromonas hydrophila* there are six LTTRs involved in the regulation of galactose metabolism. The deletion of seventh LTTRs enhanced the swarming activities of this bacterium. Furthermore, the deletion of four LTTRs resulted in increased tolerance to copper and zinc ions. Additionally, the loss of six LTTRs genes reduced the tolerance to oxidative stress [60]. Thus, the role of multiple LTTRs in specific biological process is not exclusive of *S.* Typhi, suggesting pleiotropy and potential cross-talking between LTTRs. In this regard there are other evidences of cross-talking between LTTRs. In *E. coli* it is reported that LTTRs can interact with 4, 5 or up to 10 different LTTRs [61]. These interactions may be due to the involvement of multiple LTTRs in specific biological processes [62]

LTTRs are present in Gram-negative bacteria and a majority of these proteins repress their own expression. This autoregulation appears to be necessary for the cell, since multiple LTTRs present in its active or overexpressed form can drastically inhibit bacterial growth. In this sense most of the LTTRs presented in this work are repressed in many of the conditions evaluated, since their deletion results in an enhancement of specific phenotypes such as swimming mobility and biofilm formation. This, indicates that these LTTRs are normally repressed and respond to specific conditions in the free-living or pathogenic lifestyles of *S.* Typhi. Thus, is conceivable that the *S.* Typhi LTTRs represent a repressed genetic pool indispensable in different process of *Salmonella* biology. This assertion is reinforced by the fact that other LTTRs such as LeuO are also quiescents and its role have only been observed when this protein is induced artificially [7].

LTTRs study illustrated here have extended our knowledge of their role in adverse conditions presents in the human host. Thus, LTTRs are probably fundamentals for an efficient infection. Therefore, these LTTRs deserve special attention for future studies in *S.* Typhi with the aim of understand more about bacterial pathogenesis. A more in-depth understanding of the function and contribution of *S.* Typhi LTTR determinants in infections will enable us to develop anti-virulence strategies to counteract typhoid fever.

## Supporting information

**S1 Table. Bacterial strains and plasmids used in this work.**
(DOCX)

**S2 Table. Oligonucleotides used in this work for mutants construction and transcriptional fusions.**
(DOCX)

**S3 Table. List of proteins and models used for structural alignment.**
(DOCX)

**S1 Fig. Growth curves of *S*. Typhi IMSS-1 and LTTR mutant strains in N-MM at 37°C.**
(TIF)

## Acknowledgments

We would like to thank M. Dunn for stimulating discussions and critical reading. M. Fernández-Mora, F. J. Santana and A. Vazquez for scientific suggestions. Alcocer A. and Marcos-Duran J. M. for their technical assistance.

## Author contributions

**Conceptualization:** Victor M Hernandez, Ismael Hernandez-Lucas.

**Data curation:** Victor M Hernandez, Yitzel Gama-Martínez, Lorenzo Segovia.

**Formal analysis:** Victor M Hernandez, Ismael Hernandez-Lucas.

**Funding acquisition:** Ismael Hernandez-Lucas.

**Investigation:** Victor M Hernandez, Ismael Hernandez-Lucas.

**Methodology:** Victor M Hernandez, Yitzel Gama-Martínez, Lorenzo Segovia.

**Project administration:** Ismael Hernandez-Lucas.

**Resources:** Ismael Hernandez-Lucas.

**Supervision:** Victor M Hernandez, Ismael Hernandez-Lucas.

**Validation:** Victor M Hernandez, Yitzel Gama-Martínez.

**Visualization:** Victor M Hernandez, Lorenzo Segovia, Ismael Hernandez-Lucas.

**Writing – original draft:** Victor M Hernandez, Ismael Hernandez-Lucas.

**Writing – review & editing:** Victor M Hernandez, Yitzel Gama-Martínez, Lorenzo Segovia, Ismael Hernandez-Lucas.

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
