## [Decision Letter · Decision Letter 0]

24 Oct 2025

Dear Dr. Hernandez Lucas,

Thank you for submitting your manuscript to PLOS ONE. After careful consideration, we feel that it has merit but does not fully meet PLOS ONE’s publication criteria as it currently stands. Therefore, we invite you to submit a revised version of the manuscript that addresses the points raised during the review process.

We look forward to receiving your revised manuscript.

Kind regards,

Mohammad Faezi Ghasemi, Ph.D

Academic Editor

PLOS ONE

Journal Requirements:

“This work was supported by grants from: Dirección General de Asuntos del Personal Académico (DGAPA/UNAM, IN203621-IN202224) and Consejo Nacional de Humanidades Ciencia y Tecnología (CONAHCYT, CF-2023-I-2079) to I.H.L. V. H was supported by Universidad Nacional Autónoma de México Posdoctotal Programs (POSDOC).”

“We would like to thank M. Dunn for stimulating discussions and critical reading. M. 476 Fernández-Mora, F. J. Santana and A. Vazquez for scientific suggestions. Alcocer A. 477 and Marcos-Duran J. M. for their technical assistance. This work was supported by 23 478 grants from: Dirección General de Asuntos del Personal Académico (DGAPA/UNAM, 479 IN203621-IN202224) and Consejo Nacional de Humanidades Ciencia y Tecnología 480 (CONAHCYT, CF-2023-I-2079) to I.H.L. V. H was supported by Universidad Nacional 481 Autónoma de México Posdoctoral Programs (POSDOC)”

“This work was supported by grants from: Dirección General de Asuntos del Personal Académico (DGAPA/UNAM, IN203621-IN202224) and Consejo Nacional de Humanidades Ciencia y Tecnología (CONAHCYT, CF-2023-I-2079) to I.H.L. V. H was supported by Universidad Nacional Autónoma de México Posdoctotal Programs (POSDOC).”

5. Please ensure that you refer to Figure 1, 6, 8 and 9 in your text as, if accepted, production will need this reference to link the reader to the figure.

Reviewer's Responses to Questions

**Comments to the Author**

1. Is the manuscript technically sound, and do the data support the conclusions?

Reviewer #1: Partly

Reviewer #2: Yes

2. Has the statistical analysis been performed appropriately and rigorously?

Reviewer #1: Yes

Reviewer #2: Yes

3. Have the authors made all data underlying the findings in their manuscript fully available?

Reviewer #1: No

Reviewer #2: Yes

4. Is the manuscript presented in an intelligible fashion and written in standard English?

Reviewer #1: Yes

Reviewer #2: Yes

Reviewer #1: This manuscript presents a highly comprehensive and systematic functional characterization of 18 previously uncharacterized LysR-Type Transcriptional Regulators (LTTRs) in Salmonella enterica serovar Typhi. The high-throughput approach, covering key virulence-associated phenotypes such as bile tolerance, antimicrobial resistance, motility, and biofilm formation, represents a significant contribution to S. Typhi pathogenesis research. The methodological rigor, particularly the use of non-polar mutagenesis and host-relevant minimal medium, makes this work suitable for publication in PLOS ONE, pending revisions.

Overall Assessment:

Major Revisions are suggested to enhance the mechanistic depth and structural completeness of the manuscript.

Reviewer #2: 1. The authors should clarify whether the identified LTTRs directly regulate ompC and ompF transcription or whether their effects are indirect. Likewise, for biofilm formation and motility, it would strengthen the manuscript to indicate whether specific virulence- or flagellar-related genes are misregulated in the mutants. Including even preliminary data or a focused discussion on potential regulatory targets would greatly enhance the study’s impact.

2. The assignment of mutants into four functional groups appears somewhat arbitrary and is not consistently aligned with the experimental data. For example, the manuscript states that Group II mutants “produced OmpF but not OmpC,” yet the corresponding figures suggest that some Group II strains express both porins, while others like ΔSTY0341 (also classified as Group II) lack OmpF entirely. The authors should carefully re-evaluate this classification using all available phenotypic evidence and provide a clear, data-driven justification for the final groupings.

3. The introduction would benefit from a brief mention of the clinical significance of typhoid fever and the growing challenge of antibiotic resistance. This context would more effectively underscore the relevance of investigating virulence mechanisms in Salmonella typhi.

4. In the discussion, the authors should either temper or remove the claim regarding the global regulon of the transcription factor, as 1D gel electrophoresis alone is insufficient to support such a broad conclusion. A comprehensive assessment of a regulon would require more robust approaches, such as proteomic or transcriptomic analyses.

**Do you want your identity to be public for this peer review?** For information about this choice, including consent withdrawal, please see our Privacy Policy

Reviewer #1: No

Reviewer #2: **Yes: ** Ashraf Kariminik

---

## [Author Response · Author response to Decision Letter 1]

7 Nov 2025

Responses to reviewers are in bold.

Reviewer #1: This manuscript presents a highly comprehensive and systematic functional characterization of 18 previously uncharacterized LysR-Type Transcriptional Regulators (LTTRs) in Salmonella enterica serovar Typhi. The high-throughput approach, covering key virulence-associated phenotypes such as bile tolerance, antimicrobial resistance, motility, and biofilm formation, represents a significant contribution to S. Typhi pathogenesis research. The methodological rigor, particularly the use of non-polar mutagenesis and host-relevant minimal medium, makes this work suitable for publication in PLOS ONE, pending revisions.

Overall Assessment:

Major Revisions are suggested to enhance the mechanistic depth and structural completeness of the manuscript.

Reviewer #2: 1. The authors should clarify whether the identified LTTRs directly regulate ompC and ompF transcription or whether their effects are indirect. Likewise, for biofilm formation and motility, it would strengthen the manuscript to indicate whether specific virulence- or flagellar-related genes are misregulated in the mutants. Including even preliminary data or a focused discussion on potential regulatory targets would greatly enhance the study’s impact.

Response:

Regarding the first point about porin regulation, previous work by our group has focused on the ompC and ompF regulation mediated by LTTRs, and the results demonstrate that the LTTRs STY2660 and STY0036 described in this work, directly control the expression of the master porin regulator ompR and this protein directly induces the expression of both ompC and ompF. Thus, this LTTRs are pivotal indirectly in the ompC and ompF genetic expression in S. Typhi (7, 9). This paragraph was included in the new version of the manuscript.

Regarding the second point we perform a string analysis of the 18 LTTRs described in this work and the results shows that the LTTR STY0341 characterized in this study is homologous to SinR from Salmonella Typhimurium. Previous studies have shown that sinR mutants are attenuated in their ability to replicate within macrophages, and that sinR expression is regulated by HilD, a major transcriptional activator involved in Salmonella invasion of intestinal epithelial cells (58,59). These findings suggest that STY0341 could also play a role in Salmonella Typhi virulence. Furthermore, in silico analysis using STRING (60) revealed that STY0341 is co-expressed with HypT, a protein involved in oxidative stress resistance in S. Typhimurium. In contrast, the STRING analysis of the remaining 17 LTTRs showed no co-expression with known virulence genes or with transcriptional regulators associated with Salmonella's free-living state. This paragraph was included in the new version of the manuscript.

2. The assignment of mutants into four functional groups appears somewhat arbitrary and is not consistently aligned with the experimental data. For example, the manuscript states that Group II mutants “produced OmpF but not OmpC,” yet the corresponding figures suggest that some Group II strains express both porins, while others like ΔSTY0341 (also classified as Group II) lack OmpF entirely. The authors should carefully re-evaluate this classification using all available phenotypic evidence and provide a clear, data-driven justification for the final groupings.

Response:

As you suggest a carefully revision of the section mentioned was performed and a mistake in group III regarding motility establish that motility was affected negatively. However, the correct phenotype is that motility was enhanced. The new revised version of the manuscript includes the correction.

3. The introduction would benefit from a brief mention of the clinical significance of typhoid fever and the growing challenge of antibiotic resistance. This context would more effectively underscore the relevance of investigating virulence mechanisms in Salmonella typhi.

Response:

We included a brief mention of the clinical significance of typhoid fever and the growing challenge of antibiotic resistance in the introduction as you suggested.

4. In the discussion, the authors should either temper or remove the claim regarding the global regulon of the transcription factor, as 1D gel electrophoresis alone is insufficient to support such a broad conclusion. A comprehensive assessment of a regulon would require more robust approaches, such as proteomic or transcriptomic analyses.

Response:

We are in agreement with your observation, thus we eliminated this paragraph of the discussion as you suggested.

---

## [Decision Letter · Decision Letter 1]

18 Nov 2025

Functional role of 18 LysR-Type transcriptional regulators of Salmonella enterica serovar Typhi.

PONE-D-25-53096R1

Dear Dr. Hernandez Lucas,

We’re pleased to inform you that your manuscript has been judged scientifically suitable for publication and will be formally accepted for publication once it meets all outstanding technical requirements.

Kind regards,

Mohammad Faezi Ghasemi, Ph.D

Academic Editor

PLOS ONE

Additional Editor Comments (optional):

Reviewers' comments:

Reviewer's Responses to Questions

**Comments to the Author**

Reviewer #2: All comments have been addressed

2. Is the manuscript technically sound, and do the data support the conclusions?

Reviewer #2: Yes

3. Has the statistical analysis been performed appropriately and rigorously?

Reviewer #2: Yes

4. Have the authors made all data underlying the findings in their manuscript fully available?

Reviewer #2: Yes

5. Is the manuscript presented in an intelligible fashion and written in standard English?

Reviewer #2: Yes

Reviewer #2: (No Response)

**Do you want your identity to be public for this peer review?** For information about this choice, including consent withdrawal, please see our Privacy Policy

Reviewer #2: **Yes: ** Ashraf Kariminik

---

## [Editor Report · Acceptance letter]

PONE-D-25-53096R1

PLOS ONE

Dear Dr. Hernandez-Lucas,

I'm pleased to inform you that your manuscript has been deemed suitable for publication in PLOS ONE. Congratulations! Your manuscript is now being handed over to our production team.

Kind regards,

on behalf of

Dr. Mohammad Faezi Ghasemi

Academic Editor

PLOS ONE